# Spin-orbit torque switching of an antiferromagnetic metallic heterostructure

Samik DuttaGupta [1,2,3✉], A. Kurenkov [1,2,3], Oleg A. Tretiakov [4], G. Krishnaswamy[5], G. Sala [5], V. Krizakova [5], F. Maccherozzi[6], S. S. Dhesi[6], P. Gambardella [5], S. Fukami [1,2,3,7,8] & H. Ohno[1,2,3,7,8]

The ability to represent information using an antiferromagnetic material is attractive for future antiferromagnetic spintronic devices. Previous studies have focussed on the utilization of antiferromagnetic materials with biaxial magnetic anisotropy for electrical manipulation. A practical realization of these antiferromagnetic devices is limited by the requirement of material-specific constraints. Here, we demonstrate current-induced switching in a poly-crystalline PtMn/Pt metallic heterostructure. A comparison of electrical transport measurements in PtMn with and without the Pt layer, corroborated by x-ray imaging, reveals reversible switching of the thermally-stable antiferromagnetic Néel vector by spin-orbit torques. The presented results demonstrate the potential of polycrystalline metals for antiferromagnetic spintronics.

[1] Center for Science and Innovation in Spintronics, Tohoku University, 2-1-1 Katahira, Aoba-ku, Sendai 980-8577, Japan. [2] Center for Spintronics Research Network, Tohoku University, 2-1-1 Katahira, Aoba-ku, Sendai 980-8577, Japan. [3] Laboratory for Nanoelectronics and Spintronics, Research Institute of Electrical Communication, Tohoku University, 2-1-1 Katahira, Aoba-ku, Sendai 980-8577, Japan. [4] School of Physics, The University of New South Wales, Sydney 2052, Australia. [5] Laboratory for Magnetism and Interface Physics, Department of Materials, ETH Zurich, 8093 Zurich, Switzerland. [6] Diamond Light Source, Chilton, Didcot, Oxfordshire OX11 0DE, United Kingdom. [7] Center for Innovative Integrated Electronic Systems, Tohoku University,  468-1 Aramaki Aza Aoba, Aoba-ku, Sendai 980-0845, Japan. [8] WPI Advanced Institute for Materials Research, Tohoku University, 2-1-1 Katahira, Aoba-ku, Sendai 980-8577, Japan. ✉email: sdg@riec.tohoku.ac.jp

The capability to utilize antiferromagnets (AFMs) as multifunctional components of spintronic devices has opened new avenues for future spintronic devices[1–13]. Previous works utilizing antiferromagnetic heterostructures have demonstrated promising characteristics suitable for AFM-based memories[7,8], AFM/ferromagnet (FM) spin-orbit torque (SOT) device for spintronics-based neuromorphic hardware[8,9], and skyrmion-based computing[11]. Realizing the full potential of antiferromagnetic spintronics requires AFM-based components that can complement the essential functions of existing FM-based spintronics devices. To achieve this objective, the capability to electrically record and retrieve information from an antiferromagnetic material is of paramount importance. Previous works investigated the interaction of current with the antiferromagnetic Néel vector (staggered moment) resulting in anisotropic and spin-Hall magnetoresistance (SMR) effects[14–17], which can serve as an electrical probe for reading. Switching of an AFM either by field-like Néel SOTs originating from inverse spin galvanic effects[7,18–21] or SOTs in AFM-insulator/heavy metal (HM) heterostructures[22–24] and electric field control of Néel SOTs[25] have been demonstrated, offering techniques to manipulate antiferromagnetic Néel vector. However, a practical realization of these antiferromagnetic devices relies on the requirement of materials obeying certain crystallographic or magnetic symmetries[7,18–21,26] and epitaxy of the AFM with adjacent layers of the heterostructures[22–24]. A second challenge constraining the choice of material systems for antiferromagnetic spintronics concerns the stability of recorded information. Antiferromagnetic materials possessing a high thermal stability factor ($\Delta$) could be beneficial for robust storage of information free from thermally-activated intrinsic relaxation dynamics[20,27,28]. These requirements for reading, writing, and storage of information pose a stringent set of parameters limiting the materials available for antiferromagnetic spintronics. Mn-based binary metallic alloys (ex. PtMn, IrMn, etc.) corresponds to a class of specialized material, traditionally utilized in spin-valve structures owing to its capabilities of significant exchange bias field, low processing temperatures, and compatibility with Si-based electronics. The favorable combination of room temperature ordering[29,30], high thermal stability[31], significant bulk uniaxial magnetic anisotropy[32], and magnetoresistive effects in PtMn for reading[17] renders this material feasible for future antiferromagnetic spintronic devices.

Here, we show electrical writing of information in polycrystalline AFM/HM metallic structures. Electrical measurements supplemented by x-ray magnetic dichroism imaging show a deterministic reversal of the antiferromagnetic Néel vector in the metallic AFM PtMn. A comparison of electrical measurements of antiferromagnetic heterostructures with and without HM layer clarifies the underlying role played by SOTs in switching. We also demonstrate the capability of PtMn for long-time data retention due to a high thermal stability factor, intrinsic to this material[31]. The present experimental results demonstrate the prospect of metallic AFMs for future antiferromagnetic spintronic devices.

**Sample fabrication and properties**. The stack structures are deposited by magnetron sputtering on highly resistive Si substrates with a natural oxidation layer. We utilize sub./Ta(3)/Pt(3)/ MgO(2)/Pt$_{0.38}$Mn$_{0.62}$(10 ≤ $t_{PtMn}$ ≤ 30)/Pt(5)/Ru(1) (PtMn($t_{PtMn}$)/ Pt, hereafter), where the numbers in parentheses are the nominal thicknesses in nm ((Fig. 1(a))). The obtained results are compared to reference structure sub./Ta(3)/Pt(3)/MgO(2)/Pt$_{0.38}$Mn$_{0.62}$(10 ≤ $t_{PtMn}$ ≤ 30)/Ru(1) (PtMn($t_{PtMn}$)/Ru, hereafter). The deposited films are patterned into star-shaped structures by photolithography and Ar ion milling. After fabrication, we anneal the

structures at 300 °C for 2 h. Out-of-plane x-ray diffraction (XRD) spectra indicate a textured polycrystalline orientation along the (111) direction (see supplementary Fig. S1), consistent with previous reports[31,33]. We obtain an average grain size of 10 ± 2 nm by using Scherrer's formula. Magnetization hysteresis loops (m-H loops) of annealed PtMn/Pt and PtMn/Ru blanket films show a small magnitude of areal magnetic moment (m) indicating the antiferromagnetic nature of the thin films (Fig. 1(b)). The finite m can possibly arise from minute fractions of disordered moments and/or inhomogeneous multi-domain antiferromagnetic configuration (shown later). Additional m–H measurements of annealed PtMn(10)/[Co(0.3)/Ni(0.6)]$_2$/Co(0.3)/MgO(2)/Ru(1) AFM/FM heterostructures show a shifted loop indicating the presence of exchange bias[3,34], which is further proof of antiferromagnetic ordering as shown in Fig. 1(b). The star-shaped patterned samples are characterized by electrical and optical measurements. Figure 1(c)–(f) schematically shows the sequence of electrical measurements. Electrical writing of information is achieved by sourcing pulsed currents ($I_{1,2}$) along two orthogonal directions (A(C)→B(D)) while the corresponding resistance state is read-out by measuring the transverse Hall resistance ($R_{Hall}$) of the sample using a dc current ($I_{DC}$) of much weaker amplitude than $I_{1,2}$[17]. A waiting time of 10 s is employed after each write pulse. The electrical switching measurements are supplemented by x-ray magnetic linear dichroism (XMLD) measurements performed by photoemission electron microscopy (PEEM) imaging on sub./Ta(3)/Pt(5)/Pt$_{0.38}$Mn$_{0.62}$(10)/Ru(1) structures (Pt/ PtMn(10), hereafter) before and after the injection of current pulses. The position of the Pt layer in these stacks are reversed to enhance magnetic contrast from the PtMn layer. The combination of electrical measurements and XMLD-PEEM imaging enables us to clarify the characteristics of current-induced switching in the AFM PtMn.

## Results

**Experiments on current-induced switching of AFM/HM and AFM structures**. First, we investigate the current-induced switching of AFM/HM structures under the application of current pulses of varying amplitude at a constant pulse width ($\tau_P$).

Ten $I_1$ pulses are sourced along the horizontal arm (A→ B) followed by $I_2$ pulses along the vertical arm (C→ D). We measure $R_{Hall}$ after each $I_1$ or $I_2$ pulse, enabling us to detect possible changes in Néel vector under the application of current. Figure 1g shows the results of PtMn(10)/Pt for different current amplitudes at a constant $\tau_P$. The application of $I_1$ pulse results in a high-resistance state (red-rimmed circles in Fig. 1g) while $I_2$ pulse corresponds to a low-resistance state (blue-rimmed circles in Fig. 1g). This distinct nature of $R_{Hall}$ persists for different applied pulse amplitudes irrespective of $t_{PtMn}$ (see supplementary Fig. S2a, b), demonstrating the intrinsic nature of the observed behavior. Considering $R_{Hall}$ as a measure of the averaged antiferromagnetic Néel vector, distinct reversal changes under $I_1$ and $I_2$ indicate a possibility of antiferromagnetic Néel vector switching or reorientation in our AFM/HM structure. To confirm the stability of the switched states, we monitor $R_{Hall}$ for several hours after application of $I_1$ or $I_2$ (Fig. 1h). Clear, distinguishable resistive states without any sign of relaxation are observed, indicating high thermal stability of AFM PtMn[30]. Under a macrospin approximation, the thermal stability factor ($\Delta$) of an antiferromagnetic grain is expressed as $\Delta = \frac{K_U V}{k_B T}$, where $K_U$ is the anisotropy energy density, V is the grain volume, $k_B$ is the Boltzmann constant and T is temperature. Assuming $K_U \approx 1.4 \times 10^6$ J m$^{-3}$, from previous works[32], and 10 nm grain size evaluated from XRD, we obtain $\Delta \geq 150$ at 300 K, significantly higher than most of previously studied antiferromagnetic materials (~40–60)[20–24,26,27].

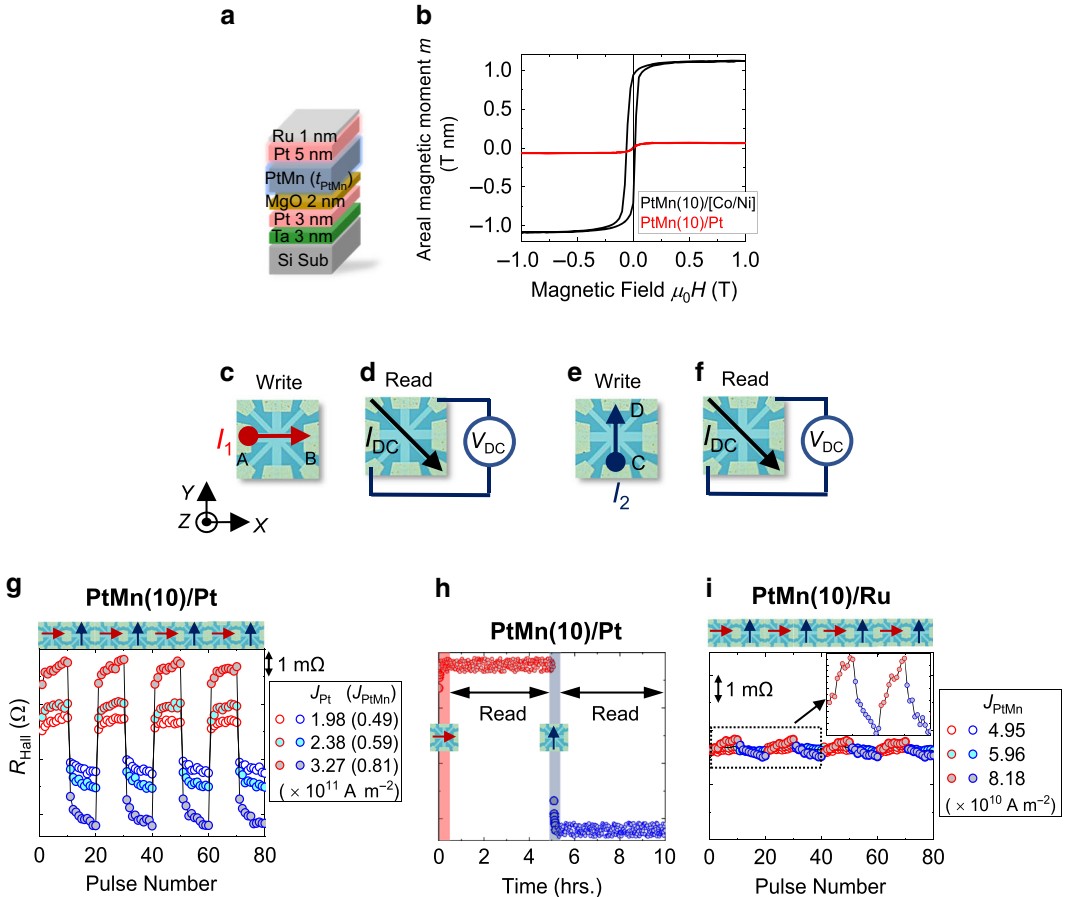

**Fig. 1 Stack structure, schematics of measurement configuration, and current-induced switching.** (**a**) Schematic diagram of the stack structure. (**b**) areal magnetic moment ($m$) vs magnetic field ($H$) for sub./PtMn(10)/[Co(0.3)/Ni(0.6)]$_2$/Co(0.3)/MgO(2)/Ru(1) (PtMn(10)/[Co/Ni]) and PtMn(10)/Pt structure. (**c**)–(**f**) Optical micrograph of the star-shaped device structure and schematic diagram of the measurement set-up. Write currents $I_1$ and $I_2$ are sourced along the paths from A(C) to B(D), respectively. Reading of the antiferromagnetic states is achieved by measuring transverse Hall voltage ($V_{DC}$) under the application of read current ($I_{DC}$) along the arm aligned at 45° to the write channel. (**g**) Experimental results of current-induced manipulation of PtMn(10)/Pt structure under applied current densities $J_{Pt} = 1.98 \times 10^{11}$ A m$^{-2}$ ($J_{PtMn} = 4.96 \times 10^{10}$ A m$^{-2}$) $J_{Pt} = 2.38 \times 10^{11}$ A m$^{-2}$ ($J_{PtMn} = 5.95 \times 10^{10}$ A m$^{-2}$) and $J_{Pt} = 3.27 \times 10^{11}$ A m$^{-2}$ ($J_{PtMn} = 8.17 \times 10^{10}$ A m$^{-2}$). (**h**) The stability of written states was investigated by measuring $R_{Hall}$ for several hours after writing. Red and blue shaded area corresponds to the writing of PtMn(10)/Pt by 10 write pulses along a direction indicated by the arrows in the schematics. The scale bar of the y-axis ($R_{Hall}$) is same as of (**g**). (**i**) Results of current-induced manipulation of PtMn(10)/Ru structure under applied $J_{PtMn} = 4.95 \times 10^{10}$, $5.96 \times 10^{10}$, and $8.18 \times 10^{10}$ A m$^{-2}$ respectively. Inset shows a magnified view of $R_{Hall}$ vs pulse number characteristics for $J_{PtMn} = 8.18 \times 10^{10}$ A m$^{-2}$ for the area bounded by the rectangular box. Schematic diagrams above (**g**) and (**i**) denotes the sequence of application of $I_1$ and $I_2$ in the respective structures.

Owing to the metallic nature of these structures, applied $I_{1,2}$ results in current flowing through both AFM and HM layers. This charge current in the HM or AFM can separately result in charge-to-spin conversion effects[35–39], both of which are capable of current-induced switching of antiferromagnetic Néel vector. To disentangle these contributions, we compare the experimental results for PtMn(10)/Pt with similar measurements on PtMn(10)/Ru at identical device dimensions (Fig. 1i). The write current density in PtMn ($J_{PtMn}$) is kept similar to that in PtMn(10)/Pt enabling a direct comparison between these two structures. As opposed to distinct reversible $R_{Hall}$ states in PtMn(10)/Pt, we observe a gradual change of $R_{Hall}$ in PtMn(10)/Ru, whose magnitude is smaller by a factor of 10 (inset of Fig. 1i). We also compare the results on PtMn(10)/Pt with similar experiments on sub./Ta(3)/Pt(5)/Ru(1) structures (see Supplementary Fig. S3a, b). No significant changes in $R_{Hall}$ are observed at comparable current densities for this structure and PtMn/Pt, ruling out any sizeable contribution from anisotropic thermo-electric effects towards the observed $R_{Hall}$ behavior[40].

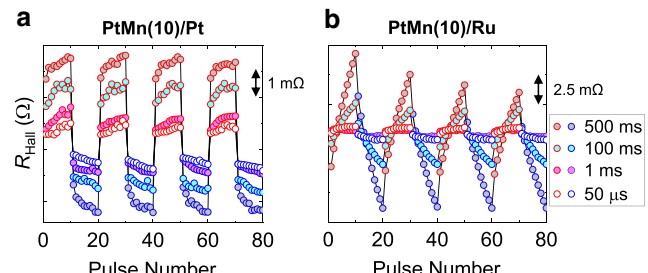

**Fig. 2 Different characteristics of current-induced switching between PtMn/Pt and PtMn/Ru structures.** (**a**) Pulse width ($\tau_P$) dependence of transverse Hall resistance ($R_{Hall}$) for PtMn(10)/Pt under applied $J_{Pt} = 3.27 \times 10^{11}$ A m$^{-2}$ ($J_{PtMn} = 8.17 \times 10^{10}$ A m$^{-2}$) for $\tau_P = 50$ μs, 1, 100 and 500 ms. (**b**) Experimental results of $\tau_P$ dependence of $R_{Hall}$ for PtMn(10)/Ru structure for applied $J_{PtMn} = 2.62 \times 10^{11}$ A/m$^2$ with similar polarities of $I_1$ and $I_2$. Red and blue-rimmed circles correspond to applied $I_1$ and $I_2$, respectively. The polarities of $I_{1,2}$ are identical to that in Fig. 1.

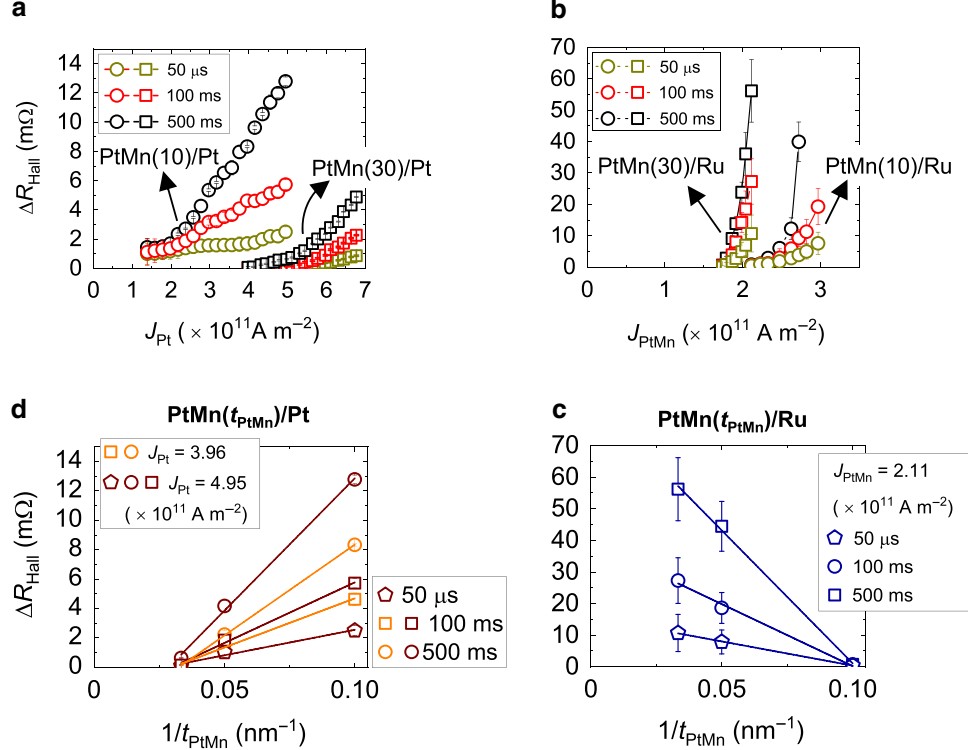

**Fig. 3 AFM thickness dependence of current-induced switching in PtMn/Pt and PtMn/Ru structures.** (**a**), (**b**) Dependence of the change in Hall resistance ($\Delta R_{Hall}$) as a function of write current densities $J_{Pt}$ and $J_{PtMn}$ for PtMn(10 or 30)/Pt and PtMn(10 or 30)/Ru structures, respectively, for various pulse widths $\tau_P$ = 50 μs, 100 ms and 500 ms. (**c**), (**d**) Inverse of AFM thickness ($t_{PtMn}$) dependence of $\Delta R_{Hall}$ for PtMn/Pt and PtMn/Ru, respectively. Solid lines in (**c**), (**d**) are guides to the eye.

To elucidate the effect of current, we then investigate $\tau_P$ dependence of switching in PtMn($t_{PtMn}$)/Pt and PtMn($t_{PtMn}$)/Ru structures. Figure 2a, b shows $\tau_P$ dependence of $R_{Hall}$ for PtMn (10)/Pt and PtMn(10)/Ru for applied $I_1$ and $I_2$ along A(C)→ B (D). For both structures, application of $I_1$ and $I_2$ pulses changes $R_{Hall}$, consistent with an interpretation of current-induced switching or reorientation of the antiferromagnetic Néel vector. A decrease of $\tau_P$ from 500 to 1 ms results in a drastic reduction of the change in $R_{Hall}$ while a further decrease to 50 μs results in a slight depreciation of $R_{Hall}$. The switching characteristics of PtMn (10)/Pt depend on current-polarity (see supplementary Fig. S4a, b) and distinctly differs from the sawtooth-like unipolar behavior in PtMn(10)/Ru (see Supplementary Fig. S5a). This sawtooth-like nature also persists for various $\tau_P$ and $t_{PtMn}$ (see supplementary Fig. S5(b)–(d)), and closely resembles the switching characteristics observed in some previous works[40,41]. The present results suggest the existence of different driving forces, manifesting in distinct $R_{Hall}$ characteristics of PtMn with/without the HM layer.

To capture the dynamics of current-induced switching, we summarize $J_{Pt}$ or $J_{PtMn}$ dependence of the change in Hall resistance ($\Delta R_{Hall}$) for PtMn(10 ≤ $t_{PtMn}$ ≤ 30)/Pt and PtMn(10 ≤ $t_{PtMn}$ ≤ 30)/Ru (Fig. 3a, b, respectively). For both the structures, an increase of write current density ($J_{Pt}$ or $J_{PtMn}$) or $\tau_P$ results in an increase of $\Delta R_{Hall}$, irrespective of $t_{PtMn}$, indicating an increased degree of antiferromagnetic Néel vector manipulation in the presence of thermal effects (as shown later). However, closer inspection reveals further differences in $\Delta R_{Hall}$ versus $J_{Pt}$ or $J_{PtMn}$ between PtMn/Pt and PtMn/Ru structures, respectively. For PtMn/Pt structures, we find evidence of two distinct regimes of $\Delta R_{Hall}$ depending on the magnitude of $J_{Pt}$; low $J_{Pt}$ regime ($J_{Pt}$ ≤ $2 \times 10^{11}$ and $5 \times 10^{11}$ A m$^{-2}$ for PtMn(10)/Pt and PtMn(30)/Pt, respectively) associated with minuscule changes in $\Delta R_{Hall}$ and a

second regime evidencing larger changes of $\Delta R_{Hall}$. For PtMn/Ru structures, unlike PtMn/Pt, we observe a sharp increase of $\Delta R_{Hall}$ confined within a small range of $J_{PtMn}$. A second difference arises in the $t_{PtMn}$ dependence of threshold current between these two structures. The threshold current required for detectable switching of the antiferromagnetic Néel vector increases with increasing $t_{PtMn}$ for PtMn/Pt structures, while showing the opposite trend for the PtMn/Ru alone. Furthermore, striking differences appear in the behavior of $\Delta R_{Hall}$ versus $1/t_{PtMn}$ for both structures for various applied write current densities and $\tau_P$ (Fig. 3c, d). An increase of $t_{PtMn}$ for PtMn/Pt results in a decrease of $\Delta R_{Hall}$ whereas again showing an opposite behavior for PtMn/Ru. As discussed later, the former trend is expected from switching of the antiferromagnetic Néel vector by SOTs from the HM layer while the latter is likely from several origins, magnetic or non-magnetic. Nevertheless, our experimental results demonstrate the possibility of current-induced switching of antiferromagnetic PtMn with relatively low current densities of ~$10^{11}$ A m$^{-2}$ from dc to μs pulses, relevant for AFM-based future spintronic devices.

**XMLD-PEEM imaging of current-induced switching in AFM/ HM structures.** To prove that the application of orthogonal write pulses indeed results in electrical switching characteristics of magnetic origin, we resort to XMLD-PEEM imaging of Pt/PtMn (10) structures. Separate electrical switching measurements confirm similar reversible current-induced switching behavior in these structures as well (see Supplementary Fig. S6). Domain imaging is carried out for both linear vertical (LV) and linear horizontal (LH) polarizations under the application of current, enabling the visualization of the magnetization configurations for high and low-resistive $R_{Hall}$ states. Figure 4a–c shows the optical micrograph of the device structure along with the directions of

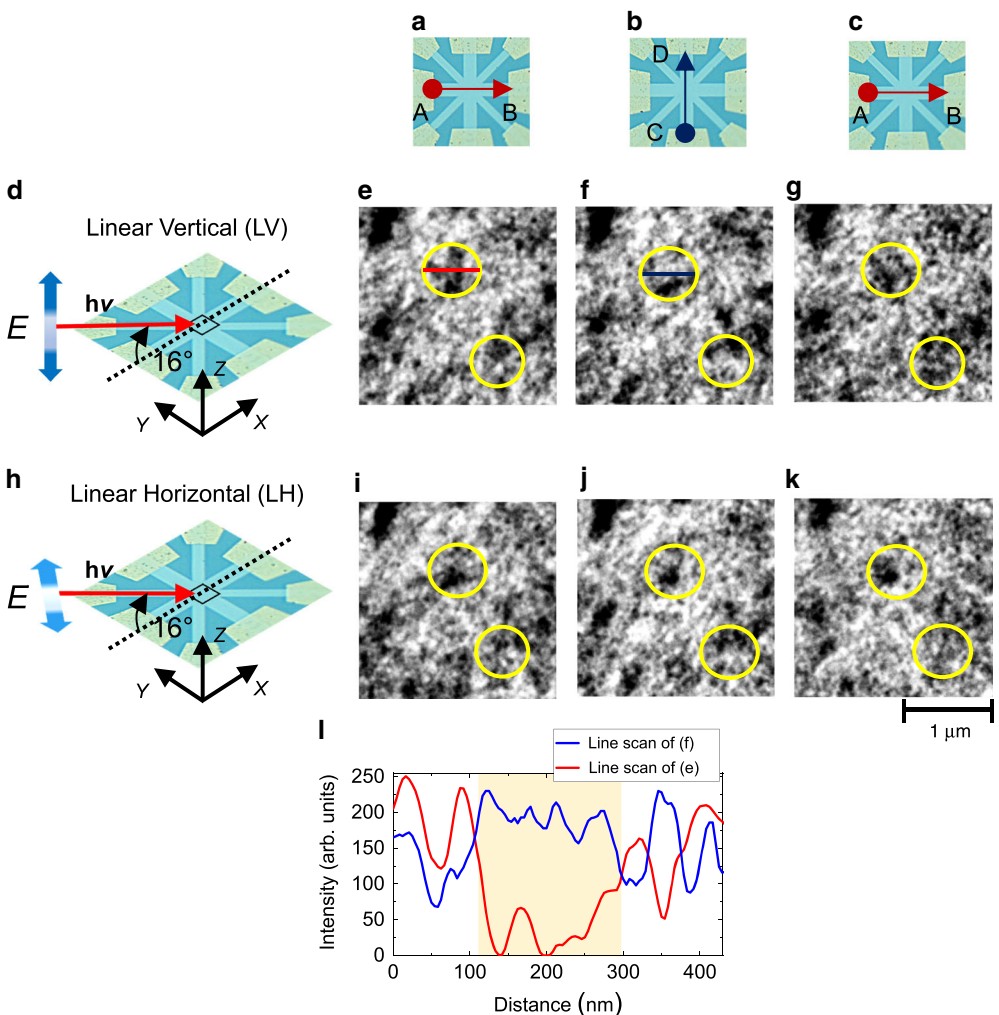

**Fig. 4 XMLD-PEEM imaging of current-induced switching in Pt/PtMn(10) structures.** (**a**)–(**c**) Schematic diagram of the sequence of applied write currents in Pt/PtMn(10). (**d**), (**h**) Schematic diagram of the measurement set-up for XMLD-PEEM imaging. X-rays are incident on the sample at an angle of 16° to the sample surface. Linear vertical (LV) and linear horizontal (LH) polarizations of the x-ray beam are indicated by thick blue arrows. The black square box of approximate size 2 μm × 2 μm at the center of the device denotes the position where the imaging was carried out. (**e**)–(**g**) LV polarization XMLD-PEEM images of Pt/PtMn(10) structure. The images were taken after injection of 20 pulses of 100 ms duration for $J_{Pt} = 5.93 \times 10^{11}$ A m$^{-2}$ along A (B)→C(D). White and black areas in the figure indicate regions with opposite linear dichroism contrast for the LV polarization of the incident beam. Yellow circles highlight regions of the sample with prominent switching. (**i**)–(**k**) LH polarization XMLD-PEEM images at the same position as (**e**)–(**g**) after the application of current pulses. White and black areas in the figure indicate regions with opposite linear dichroism contrast for the LH polarization of the incident beam. Changes due to current pulsing are visible also in these images. (**l**) Line scan of pixel intensity (in arb. unit) vs distance for red and blue lines in panels (**e**) and (**f**), respectively. The switchable antiferromagnetic domain size under the action of the current is determined from the length of the yellow shaded region.

applied $I_{1,2}$ while Fig. 4d, h shows the schematic diagram of the measurement configuration. The linear polarization of the x-rays in the LH mode is in-plane, whereas that of the LV mode makes an angle of 16° with respect to the sample normal. Figure 4e–g, i–k shows the normalized XMLD images in LV and LH mode, respectively, after the injection of orthogonal write pulses of magnitude $J_{Pt} = 5.93 \times 10^{11}$ A m$^{-2}$ and $\tau_P = 100$ ms. The XMLD asymmetry is obtained by subtracting images on and off the $L_3$ edge for each polarization (see Methods). Separate X-ray circular magnetic dichroism (XMCD) measurements at Mn $L_3$ edge on AFM/FM PtMn/[Co/Ni] multilayer structures do not reveal any discernable XMCD signals upon the application of current, ruling out any dominant contribution from disordered or uncompensated moments. Despite the possible presence of chemical and morphological contrast in both LV and LH configurations, which can be caused by variations in the stoichiometry and orientation of different crystal grains, we observe contrast reversal

in several areas (black to white, and vice-versa) following current-induced switching. As seen in Fig. 4e–g, the application of orthogonal $I_{1,2}$ results in reversible changes, which we attribute to the reversal of the antiferromagnetic Néel vector. In addition, some areas do not show any contrast reversal either in LH or LV polarization, indicating non-switchable portions, as well. Precise identification of the dynamics of the antiferromagnetic Néel vector during electrical switching requires XMLD-PEEM imaging for various orientations of the sample with respect to the incident x-rays direction, which will be investigated in future. Figure 4l shows the line scans through the LV images in Fig. 4e, f. Discernible changes in antiferromagnetic Néel vector occur over a localized region with an upper limit of sizes in the range of hundreds of nm (shaded area in Fig. 4l), consistent with previous studies using different collinear antiferromagnetic structures[42,43]. We believe that the observed current-induced switching behavior is a universal feature and offers the possibility to utilize

polycrystalline metallic AFMs within the arena of antiferromagnetic spintronics.

## Discussion

As presented above, we demonstrate current-induced 90° switching of the antiferromagnetic Néel vector in metallic AFM heterostructures. To understand the scenarios responsible for the observed switching behavior, we first address the issue of spin structure and magnetocrystalline anisotropy of PtMn. In bulk AFM crystals, PtMn has uniaxial anisotropy, which does not favor 90° switching. Our sputtered films, however, have a polycrystalline structure. Previous studies pointed out that significant magnetostriction coefficient[44] and the sensitivity of Mn-based AFMs to crystallinity and/or chemical composition[29,30,32,45] (e.g., effects of Mn substitution/doping and valence electron number) could induce an easy-plane magnetic anisotropy[46–48], resulting in multiple stable Néel vector orientations in the polycrystalline films. Note that our observation of a reversible XMLD contrast along LV and LH configurations (Fig. 4(e)–(k)) is also consistent with the above scenario, indicating the possibility of both out-of-plane and in-plane Néel vector components. Besides, to quantify the effect of interfacial chiral interactions[48,49] (e.g., Dzyaloshinskii-Moriya interaction (DMI)), we also estimate the DMI constant ($D$) (see supplementary Fig. S7). The estimated $D$ is close to the threshold value[48] (see supplementary table T1) required for the generation of an inhomogeneous ground state with Néel-type domain wall (DW)[24,49–51], and/or other topological spins textures[52–54]. The DW width, estimated to be a few nm, is much smaller than the magnetic domain or crystallographic grain size (see Supplementary Fig. S8), indicating the feasibility of the above scenario. For a polycrystalline textured PtMn, these estimates intuitively imply the possible existence of an inhomogeneous multi-domain antiferromagnetic ground state configuration with a partial or dominant easy-plane magnetic anisotropy contribution, accounting for the observed 90° switching.

Next, we discuss the effect of possible interactions namely, the bulk or interfacial SOTs generated by spin-Hall effect in HM layer[22–24,26], Néel SOTs specific to the AFM[37–39], spin-transfer torque (STT) generated by spin-polarized conduction electrons in the AFM layer[55,56], and thermal activation of antiferromagnetic grains[20,41] due to the effect of Joule heating. Owing to the negligible current flow through the oxidized Ru capping layer, we do not consider its contribution for both PtMn/Pt and PtMn/Ru structures. The lack of structural inversion asymmetry in this collinear AFM is incompatible with the description of current-induced switching by staggered field-like Néel SOTs, originating from inverse spin galvanic effects, previously demonstrated in biaxial CuMnAs and $Mn_2Au$ antiferromagnetic structures[7,8,18–21]. This rule out any presence of Néel SOTs, leading to the observed current-induced switching in both PtMn/Pt and PtMn/Ru structures. To understand the possible role of thermally-activated dynamics, we also quantify the temperature rise in PtMn/Pt and PtMn/Ru structures by using the device resistance as a thermometer probe (see supplementary Fig. S9a–d). The estimated rise in temperature of PtMn/Ru at maximum $J$ is higher than 300 K (see supplementary Fig. S9b), whereas it reaches only ~100 K in PtMn/Pt (see supplementary Fig. S9d). This substantial increase of temperature in the PtMn/Ru structures is also accompanied by localized darkening/hot spots (see supplementary Fig. S10) and a non-negligible variation of $\Delta R_{Hall}$ from sample to sample. For PtMn/Ru, the current flowing through the AFM layer can also result in STTs acting on the multi-domain ground state. Theoretical calculations have shown that adiabatic and non-adiabatic components of STTs can lead to translational motion of AFM DWs[55,56], which could result in a magnetoresistive response owing to the finite magnetization in the DW region. For this scenario, the threshold current required to induce DW motion should remain unchanged with increasing $t_{PtMn}$, roughly consistent with our observed results (Fig. 3b). However, the non-saturating behavior of switching amplitude and its variation in successive cycles indicate intermingled contributions from other sources such as thermally activated reorientation of antiferromagnetic Néel vector[20,41] and/or non-magnetic contributions arising from electromigration[40]. Whereas our electrical measurements do not enable us to distinguish these factors, the present observations are likely related to a combination of thermally-activated reorientation of the antiferromagnetic Néel vector with possible contributions from STT and other non-magnetic effects. On the other hand, the opposite dependence of $\Delta R_{Hall}$ versus $1/t_{PtMn}$ for PtMn/Pt structures and a much smaller temperature rise suggest a different scenario. Owing to the significantly lower switching amplitude of PtMn/Ru as compared to PtMn/Pt at comparable $J_{PtMn}$ (Fig.1g, i), we ignore any contributions from STTs, and only consider the effects of bulk or interfacial SOTs on the inhomogeneous multi-domain antiferromagnetic ground state for a qualitative understanding of the switching behavior. Note that the current-polarity dependent switching in our PtMn/Pt structures is significantly different from the unipolar characteristics in PtMn/Ru and to those previously observed in AFM-insulator NiO/HM structures[22–24,57], calling for additional factors leading to the observed results. As stated before, a combination of uniaxial and easy-plane anisotropies along with interfacial DMI can lead to the spontaneous multi-domain configuration comprising Néel DWs and/or topological spin textures[47,48,52–54] in PtMn/Pt structures. In fact, these predictions have been confirmed by recent experiments demonstrating imprinted antiferromagnetic vortex states on an adjacent ferromagnetic layer in AFM/FM[52,53] or exotic topological meron-antimeron pairs in AFM/HM structures[54]. The twisting of the antiferromagnetic Néel vector around these spin textures leads to non-zero Néel topological charge, endowing protected spin configurations with distinct magnetic polarities and chiralities (sizes ~ hundreds of nm) and robust thermal stability. Besides, numerical simulations also suggest efficient nucleation and motion of these antiferromagnetic spin textures under the action of SOTs in AFM/HM[58]. Thus, a possible scenario concerning the polarity dependent current-induced switching characteristics is attributed to the action of SOTs on the inhomogeneous multi-domain configuration. Irrespective of the initial multi-domain configuration, the switching dynamics is expected to proceed via a 90° rotation of the in-plane projection of Néel vector towards the spin-polarization direction[48] by rearrangement or motion of DWs, and/or current-induced nucleation and motion of vortex-antivortex pairs, skyrmions, or bimerons[54,58]. An increase of the switching amplitude ($\Delta R_{Hall}$) versus $J_{Pt}$ is attributed to enhanced current-induced nucleation or rearrangement of these spin textures, whereas the partial switching might be due to pinning effects. In principle, the threshold-like behavior observed in $\Delta R_{Hall}$ versus $J_{Pt}$, can be attributed to the cross-over between various regimes of spin texture dynamics characterized by different pinning strengths. Furthermore, an increase of $t_{PtMn}$ above the spin-diffusion length results in decreased efficiencies of SOTs and lower probability of switching, similar to the ferromagnetic case. Contrary to the common understanding that biaxial AFMs are required for electrical switching, our results demonstrate the potential of polycrystalline antiferromagnetic metals compatible with existing complementary metal-oxide semiconductor (CMOS) technology for low current operation and high thermal stability antiferromagnetic spintronics. We also believe that the present results indicate an intricate role played by topological

spin textures for current-induced switching and points towards an unexplored pathway for their utilization in future anti-ferromagnetic spintronic devices.

In summary, we demonstrate SOT-induced switching of the Néel vector in a Mn-based metallic AFM/HM heterostructure at low current densities of $\approx 10^{11}$ A m$^{-2}$ from dc to µs regime. The comparison of the electrical measurements in AFM structures with and without the HM layer allows us to distinguish different current-induced effects in AFMs. The combination of electrical measurements with x-ray imaging clearly shows reversible switching characteristics of the antiferromagnetic Néel vector within localized regions of sizes of hundreds of nm. The large $\Delta$ for the antiferromagnetic PtMn ensures a stable Néel vector orientation, implying robust data retention capabilities of this material system. The present study shows that polycrystalline metallic AFM/HM structures are promising candidates for anti-ferromagnetic spintronics.

## Method

**Film preparation**. The films were deposited at room temperature onto 3-inch high-resistive Si wafers with a natural oxidation layer. RF magnetron sputtering was used to deposit the MgO layer, and DC magnetron sputtering was used to deposit the other layers. Base pressure of the chamber was less than $1 \times 10^{-6}$ Pa, and Ar gas was used for sputtering. No magnetic field was applied during the sputtering. The composition of PtMn sputtering target is Pt$_{38}$Mn$_{62}$ (in atomic %).

**Device fabrication**. The deposited films of Ta(3)/Pt(3)/ MgO(2)/PtMn($t_{PtMn}$)/Pt (5)/Ru(1) and Ta(3)/Pt(3)/ MgO(2)/PtMn($t_{PtMn}$)/Ru(1) were processed into star-shaped devices by photolithography and Ar ion milling. Electrodes and contact pads made of Cr (5 nm)/Au (100 nm) were formed by photolithography and lift-off. Width and length of the write channel of the star-shaped devices were 10 and 60 µm, respectively, and those of the read channel were 5 and 60 µm, respectively. After the fabrication, the samples were annealed at 300 °C for two hours under an in-plane magnetic field of 1.2 T directed parallel to the write channel. Resistivities of Ta, Pt, and Pt$_{0.38}$Mn$_{0.62}$ layers were determined from separate measurements of sheet resistance on annealed blanket films and is reported elsewhere[17].

**XMLD-PEEM measurements**. X-ray absorption spectroscopy and XMLD-PEEM experiments, with 10 µm field of view, were performed at the I06 beamline at the Diamond Light Source, UK. PEEM images were acquired with x-ray energies at $E_1 = 638.15$ eV and $E_2 = 638.95$ eV, incident at an angle of 16° on the sample surface (schematic of the measurement is shown in Fig. 4). $E_2$ and $E_1$ correspond to the energies closer to the Mn $L_3$ edge at which the maximum XMLD contrast is obtained, resulting in a difference in absorption between regions with spin-axis collinear or perpendicular to the incident x-ray polarization[59]. XMLD images were obtained by calculating the normalized difference of consecutive images measured at $E_{1,2}$, that is, by calculating the intensity of each pixel as $[i(E_1) - i(E_2)]/[i(E_1) + i(E_2)]$. The images were acquired for both linear vertical (LV) and linear horizontal (LH) polarization of the incident x-rays, before and after the application of current pulses. Note that, in addition to the XMLD effect, the intensity asymmetry calculated by subtracting images on and off the $L_3$ edge also depends on local variations of the x-ray absorption intensity due to the different stoichiometry and orientation of the PtMn crystal grains. Contrast changes following the application of current pulses, however, are purely of magnetic origin.

## Data availability

The data which support the findings of this work are available from the corresponding author upon reasonable request.

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

## Acknowledgments

We thank C. Igarashi, T. Hirata, H. Iwanuma, and K. Goto for their technical support and Dr. J. Llandro for discussions. A portion of this work was supported by the JSPS KAKENHI 17H06511, 18KK0143, 19H05622, and Cooperative Research Projects of RIEC. O.A.T. acknowledges support by the Australian Research Council (Grant No. DP200101027), the Cooperative Research Project Program at the Research Institute of Electrical Communication, Tohoku University, and by the NCMAS 2020 grant. G.K., G.S., and P.G. acknowledge funding from the Swiss National Science Foundation (Grant No. 200020-172775). V.K. was supported by a Swiss Government Excellence Scholarship.

## Author contributions

S.D. and S.F. planned the study. S.D. deposited the films and fabricated the samples. S.D. performed the electrical measurements and analyzed the data with inputs from A.K. and O.A.T. A.K., G.K., G.S., V.K., and F.M. measured and analyzed the XMLD data with input from P.G. S.D. wrote the manuscript with input from all the authors. All authors discussed the results.

## Competing interests

The authors declare no competing interests.
