## [Peer Review File · Nature Communications]

Reviewers' Comments:

Reviewer #1:

Remarks to the Author:

The paper demonstrates possibility of all electrical manipulation of the device based on an antiferromagnet PtMn which can find applications for information coding. From comparison of different devices (with and without heavy metal) the authors show that the switching is induced by SOT (from Pt electrode) and that antiferromagnetic ordering of the active layer plays major role in the switching process. Experimental study is very detailed and includes not only electrical, but also XMLD measurements which give direct access to the magnetic ordering in the device. The paper is written in a clear, transparent manner and can have a great impact on the spintronic society. I believe that these results should be published, and should be published in high-rank journal. However, I found theoretical interpretation rather vague and would suggest to clarify some points before publication.

1. The authors believe that magnetic structure of an AFM is uniaxial, as concluded from the previous studies for bulk sample. However, all the technique used for detection of switching (SMR, XMLD) is sensitive to 90° switching which is inconsistent with uniaxial symmetry. Moreover, the authors demonstrate high thermal stability of TWO states with different resistivity. In a simple picture of single-particle switching there is only one physically resolved state (in case of uniaxial symmetry). If the picture is not so simple, this should be explained within the main text, to avoid misleading.

2. In assumption of uniaxial symmetry, SOT-induced motion of 180° domain walls (suggested as one of the possible mechanism) should not create disbalance between the domains, as the material (PtMn) show no DMI. Moreover, the reported grain size (10nm) seems to be below single-domain limit which excludes mechanisms based on domain wall motion. In any case, estimation of the domain wall width and comparison with the grain size would be helpful.

3. Line 266 "This rule out the presence of relativistic SOTs". I guess that the authors mean the Néel SOT. All SOTs are by definition relativistic, as they are based on spin-orbit interactions.

4. The word "triangular" in application to the pulse shape is used throughout the text. I am not a purist, but mathematically triangle is a closed figure, in contrast to a signal which is a function of time.

To conclude: I would recommend to reconsider after revision.

Reviewer #2:

Remarks to the Author:

This manuscript reports an experiment on current-induced switching of antiferromagnetic order in PtMn. Two different structures, PtMn/Pt and PtMn/Ru, were compared. Several qualitative differences in the switching behavior between two structures were observed. (i) The variation of R_{Hall} versus pulse number is different (Fig. 2): more or less rectangle (PtMn/Pt) versus triangular (PtMn/Ru). (ii) ΔR_{Hall} versus current density is different (Fig. 3a and 3b): two thresholds (PtMn/Pt) versus a single threshold or unclear (PtMn/Ru). (iii) Dependence of ΔR_{Hall} on $1/(\text{PtMn thickness})$ is different (Fig. 3c and 3d): proportional (PtMn/Pt) and inversely proportional (PtMn/Ru). Unfortunately, the manuscript does not provide reasonable explanations for these differences. An additional XMCD measurement was carried out, which supports that the antiferromagnetic order is indeed switched by current injection. At the same time, the XMCD result shows that the current-induced switching is irregular and partial.

Given the recent considerable interest in antiferromagnetic spintronics, an experimental demonstration of practically useful current-induced switching of antiferromagnetic order would be

important. Previous experiments on antiferromagnets with biaxial anisotropy found that the switching is partial and highly random. PtMn with uniaxial anisotropy, investigated in this work, also shows partial and random switching. In this respect, this work fails to evidence that uniaxial antiferromagnets are superior to biaxial ones for switching devices. Moreover, even though the authors observed several interesting features listed above, they do not provide any clear explanations. For instance, concerning (ii), no explanation is given for two thresholds observed in PtMn/Pt. Moreover, concerning (iii), the authors explained the difference based on different Joule heating. If the current-induced variation of R_{Hall} of PtMn/Ru originates entirely from Joule heating, why does R_{Hall} of PtMn/Ru increase or decrease depending on the current direction (i.e., A-B vs C-D)? There are many other open questions in this work, which I believe must be answered at least partially.

Some minor questions:

1. It seems that Ru(1 nm) is used as a capping layer of PtMn/Ru. Does it free from oxidation when the sample is exposed to air? If oxidized, does it affect the spin-orbit torque?
2. In Fig. 1(b), the magnetic moment of PtMn/Pt is small but clearly finite. Where does it come from?
3. What is the easy axis of PtMn? Is it normal to the plane?

Overall, I do not find that this manuscript deserves publication in Nature Communications for the above reasons.

Reviewer #3:

Remarks to the Author:

The manuscript describes a spin orbit torque induced magnetization switching in antiferromagnet PtMn layer at room temperature and the PtMn is uniaxial. The work is potentially interesting and present that a uniaxial antiferromagnets could also be switched via spin orbit torque. They also provide XMLD-PEEM images to support the AFM moments switching in PtMn.

Based on the above, I feel that the work described fits within the scope of Nature Communications. Nevertheless I find some of the explanations a bit obscure and the following points must be addressed.

1. My main issue is the justification of magnetic anisotropy of PtMn. The authors mentioned the bulk PtMn occupy a uniaxial anisotropy. However, the PtMn in this manuscript is polycrystalline, which indicates the PtMn might not be uniaxial anymore. Could the author show the XRD spectra of PtMn? Whether the SOT switching could happen in highly textured PtMn?
2. The AFM moments dynamics is not clear in Fig.4. Whether the AFM moments are switched perpendicular to the spin polarization direction(parallel to current)?
3. The switching behavior in PtMn doesn't resemble the characteristics of Mn₂Au as it is nearly saturated in Mn₂Au cases.
4. The authors observe a current-polarity dependent switching behaviors. I'd like to see more discussions on this issue.

We are indebted to the reviewers for their careful readings and insightful comments, which have been valuable for us to improve the manuscript substantially.

Reviewer #1 (Remarks to the Author)

Reviewer's Comment

The paper demonstrates possibility of all electrical manipulation of the device based on an antiferromagnet PtMn which can find applications for information coding. From comparison of different devices (with and without heavy metal) the authors show that the switching is induced by SOT (from Pt electrode) and that antiferromagnetic ordering of the active layer plays major role in the switching process. Experimental study is very detailed and includes not only electrical, but also XMLD measurements which give direct access to the magnetic ordering in the device. The paper is written in a clear, transparent manner and can have a great impact on the spintronic society. I believe that these results should be published and should be published in high-rank journal. However, I found theoretical interpretation rather vague and would suggest to clarify some points before publication.

Our reply

We are pleased to learn that the reviewer has recognized the novelty and importance of our work and support the publication of our manuscript. We also thank the reviewer for summarizing the critical points of our work.

Reviewer's Comment

The authors believe that magnetic structure of an AFM is uniaxial, as concluded from the previous studies for bulk sample. However, all the technique used for detection of switching (SMR, XMLD) is sensitive to 90° switching which is inconsistent with uniaxial symmetry. Moreover, the authors demonstrate high thermal stability of TWO states with different resistivity. In a simple picture of single particle switching there is only one physically resolved state (in case of uniaxial symmetry). If the picture is not so simple, this should be explained within the main text, to avoid misleading.

Our Reply

We thank the reviewer for his/her question. As the reviewer pointed out, if the anisotropy is solely uniaxial without any additional effects, 90° switching is not expected. We speculate that the influence of easy-plane anisotropy contribution [R1-R3] along with interfacial chiral Dzyaloshinskii-Moriya interaction (DMI) [R3, R4] result in stabilization of inhomogeneous multi-domain antiferromagnet (AFM) ground state with Néel domain walls (DWs) [R3, R5] and/or other topological spin textures [R4, R6-R8], which are responsible for the observed 90° switching behavior. Previous first-principles calculations, including spin-orbit interactions, have shown that strain [R1] and/or chemical composition (e.g. effects of Mn

substitution/doping on band filling and density of states around Fermi level) [R2, R9] can result in the stabilization of easy-plane anisotropies in polycrystalline films, different from its bulk uniaxial counterpart. Besides, our estimate of DMI constant (D) in AFM/heavy-metal (HM) PtMn/Pt interface is comparable to the critical DMI constant (D_C) for an inhomogeneous ground state, indicating the possibility of a multi-domain configuration with virtually equivalent free energies. A detailed discussion and estimates of D and D_C using micromagnetic parameters typical for Mn-based metallic AFMs have been newly added in supplementary information S7 and S8 of the revised manuscript. Under this situation, the observed reversible resistive states in PtMn/Pt can be attributed to a reorientation of average in-plane projection of the antiferromagnetic Néel vector by rearrangement or motion of Néel DWs and/or current-induced nucleation and motion of spin textures under the action of current-induced torques. In view of these insights, we have thoroughly revised the discussion section.

The revised portion to respond to this issue is as follows:

<page 14, line 272>

As presented above, we demonstrate current-induced 90° switching of the antiferromagnetic Néel vector in metallic AFM heterostructures. To understand the scenarios responsible for the observed switching behaviour, we first address the issue of spin structure and magnetocrystalline anisotropy of PtMn. In bulk AFM crystals, PtMn has uniaxial anisotropy, which does not favour 90° switching. Our sputtered films, however, have a polycrystalline structure. Previous studies pointed out that significant magnetostriction coefficient⁴³ and the sensitivity of Mn-based AFMs to crystallinity and/or chemical composition^{28,29,31,44} (e.g. effects of Mn substitution/doping and valence electron number) could induce an easy-plane magnetic anisotropy⁴⁵⁻⁴⁷, resulting in multiple stable Néel vector orientations in the polycrystalline films. Note that our observation of a reversible XMLD contrast along LV and LH configurations (Figs. 4(e)-(k)) is also consistent with the above scenario, indicating the possibility of both out-of-plane and in-plane Néel vector components. Besides, to quantify the effect of interfacial chiral interactions^{47,48} (e.g. Dzyaloshinskii-Moriya interaction (DMI)), we also estimate the DMI constant (D) (see supplementary Fig. S7). The estimated D is close to the threshold value⁴⁷ (see supplementary table T1) required for the generation of an inhomogeneous ground state with Néel-type domain wall (DW)^{25,48-50}, and/or other topological spin textures⁵¹⁻⁵³. The DW width, estimated to be a few nm, is much smaller than the magnetic domain or crystallographic grain size (see supplementary Fig. S8), indicating the feasibility of the above scenario. For a polycrystalline textured PtMn, these estimates intuitively imply the possible existence of an inhomogeneous multi-domain antiferromagnetic ground state configuration with a partial or dominant easy-plane magnetic anisotropy contribution, accounting for the observed 90° switching.

Reviewer's Comment:

2. In assumption of uniaxial symmetry, SOT-induced motion of 180 domain walls (suggested as one of the possible mechanisms) should not create disbalance between the domains, as the material (PtMn) show no DMI. Moreover, the reported grain size (10nm) seems to be below single-domain limit which excludes mechanisms based on domain wall motion. In any case, estimation of the domain wall width and comparison with the grain size would be helpful.

Our Reply:

We thank the reviewer for his/her question. We agree with the reviewer that in the presence of only a uniaxial spin orientation without bulk DMI, the spin-orbit torque (SOT)-induced motion of 180° DWs should not produce any magnetoresistive effect or reversible current-induced changes in X-ray linear magnetic dichroism (XMLD). As mentioned above, we speculate that a combination of easy-plane anisotropy [R1, R2] and interfacial DMI [R3] in PtMn/Pt and PtMn/Ru structures results in the stabilization of an inhomogeneous multi-domain ground state with Néel DWs [R4, R5] and/or other topological spin textures [R6-R8], whose chirality is fixed by the sign and magnitude of D . Note that the observation of a finite XMLD signal in both linear vertical (LV) and linear horizontal (LH) configurations, consisting of alternating white and black areas is in qualitative agreement with the above scenario. Besides, we also calculate the static DW width (δ) of a two sublattice collinear AFM given by [R3]

$$\delta = \sqrt{\frac{2A_1 - A_2}{2K_U}} \quad (1)$$

where $A_1 (> 0)$ corresponds to the inhomogeneous component of intra sublattice exchange constant, $A_2 (< 0)$ is the inhomogeneous component of inter sublattice exchange constant, and K_U is the anisotropy constant. Figure R1 shows the estimated values of δ for various A_1 , A_2 and K_U , typical for Mn-based AFMs [R8-R10]. Clearly, δ is smaller than the crystallite size and experimentally observed domain size (~ 100 - 200 nm). These estimates of δ using micromagnetic parameters typical for Mn-based metallic AFMs [R8, R10, R11] have been newly added in supplementary information S9 of the revised manuscript.

Fig R1: (a) Estimate of antiferromagnet (AFM) domain wall (DW) width (δ) versus $|A_2|$ for $|A_1| = 0.01, 1$ and 10 pJ m^{-1} for $K_U = 10^6 \text{ J m}^{-3}$. (b) Variation of δ versus $|A_1|$ for $|A_2| = 0.01, 1$ and 10 pJ m^{-1} for $K_U = 10^6 \text{ J m}^{-3}$. (c) Variation of δ versus K_U for different values of A_1 and A_2 . The black arrow in the figures indicates the mean crystallite size obtained from X-ray diffraction measurements.

Accordingly, we have added the following sentence in our revised manuscript:

<page 15, line 287>

The DW width, estimated to be a few nm, is much smaller than the magnetic domain or crystallographic grain size (see supplementary Fig. S8), indicating the feasibility of the above scenario.

In light of these above estimates, one possible scenario for the observed switching behavior corresponds to the of the 90° rotation of the in-plane component of antiferromagnetic Néel vector between different multi-domain configurations achieved by rearrangement or SOT-induced motion of Néel DWs and/or current-induced nucleation and motion of other topological spin textures [R4-R7]. Accordingly, we have added the following sentences in our revised manuscript:

<page 16, line 329>

As stated before, a combination of uniaxial and easy-plane anisotropies along with interfacial DMI can lead to the spontaneous multi-domain configuration comprising Néel DWs and/or topological spin textures^{46,47,51-53} in PtMn/Pt structures. In fact, these predictions have been confirmed by recent experiments demonstrating imprinted antiferromagnetic vortex states on an adjacent ferromagnetic layer in AFM/FM^{51,52} or exotic topological meron-antimeron pairs in AFM/HM structures⁵³. The twisting of the antiferromagnetic Néel vector around these spin textures leads to non-zero Néel topological charge, endowing protected spin configurations with distinct magnetic polarities and chiralities (sizes ~ hundreds of nm) and robust thermal stability. Besides, numerical simulations also suggest efficient nucleation and motion of these antiferromagnetic spin textures under the action of SOTs in AFM/HM⁵⁷. Thus, a possible scenario concerning the polarity dependent current-induced switching characteristics is attributed to the action of SOTs on the inhomogeneous multi-domain configuration.

Reviewer's Comment

Line 266 "This rule out the presence of relativistic SOTs". I guess that the authors mean the Néel SOT. All SOTs are, by definition, relativistic, as they are based on spin-orbit interactions.

Our Reply:

We thank the reviewer for pointing it out. Accordingly, we have revised the manuscript as

From:

<page 12, line 259>

This rule out the presence of relativistic SOTs leading to the observed current-induced switching in PtMn/Pt and PtMn/Ru structures.

To:

<page 15, line 301>

This rule out the presence of Néel SOTs, leading to the observed current-induced switching in PtMn/Pt and PtMn/Ru structures.

Reviewer's Comment

The word "triangular" in application to the pulse shape is used throughout the text. I am not a purist, but mathematically triangle is a closed figure, in contrast to a signal which is a function of time.

Our Reply:

We thank the reviewer for this suggestion. Accordingly, we have revised the manuscript as

From:

<page 8, line 162>

The switching characteristics of PtMn(10)/Pt depends on current-polarity (see supplementary Fig. S3(a), (b)) and distinctly differs from the triangle-like unipolar behaviour in PtMn(10) (see supplementary Fig. S4(a)). This triangular switching behaviour also persists for various τ_P and t_{PtMn} (see supplementary Fig. S4(b)-(d)), and closely resembles the switching characteristics observed in some previous works^{17,19-21}.

To:

<page 10, line 184>

The switching characteristics of PtMn(10)/Pt depend on current-polarity (see supplementary Fig. S4(a), (b)) and distinctly differs from the sawtooth-like unipolar behaviour in PtMn(10) (see supplementary Fig. S5(a)). This sawtooth-like nature also persists for various τ_P and t_{PtMn} (see supplementary Fig. S5(b)-(d)), and closely resembles the switching characteristics observed in some previous works^{39,40}.

Reviewer #2 (Remarks to author)

Reviewer's Comment:

This manuscript reports an experiment on current-induced switching of antiferromagnetic order in PtMn. Two different structures PtMn/Pt and PtMn/Ru were compared. Several qualitative differences in the switching behavior between two structures were observed. (i) The variation of R_{Hall} versus pulse number is different (Fig. 2): more or less rectangle (PtMn/Pt) versus triangular (PtMn/Ru). (ii) Delta R_{Hall} versus current density is different (Fig. 3a and 3b): two thresholds (PtMn/Pt) versus a single threshold or unclear (PtMn/Ru). (iii) Dependence of Delta R_{Hall} on $1/(\text{PtMn thickness})$ is different (Fig. 3c and 3d): proportional (PtMn/Pt) and inversely proportional (PtMn/Ru). Unfortunately, the manuscript does not provide reasonable explanations for these differences. An additional XMCD measurement was carried out, which supports that the antiferromagnetic order is indeed switched by current injection. At the same time, the XMCD result shows that the current-induced switching is irregular and partial. Given the recent considerable interest in antiferromagnetic spintronics, an experimental demonstration of practically useful current-induced switching of antiferromagnetic order would be important. Previous experiments on antiferromagnets with biaxial anisotropy found that the switching is partial and highly random. PtMn with uniaxial anisotropy, investigated in this work, also shows partial and random switching. In this respect, this work fails to evidence that uniaxial antiferromagnets are superior to biaxial ones for switching devices. Moreover, even though the authors observed several interesting features listed above, they do not provide any clear explanations. For instance, concerning (ii), no explanation is given for two thresholds observed in PtMn/Pt. Moreover, concerning (iii), the authors explained the difference based on different Joule heating. If the current-induced variation of R_{Hall} of PtMn/Ru originates entirely from Joule heating, why does R_{Hall} of PtMn/Ru increase or decrease depending on the current direction (i.e., A-B vs C-D)? There are many other open questions in this work, which I believe must be answered at least partially.

Our Reply:

We thank the reviewer for summarizing the main results of our work and for his/her questions. We are also pleased to learn that the reviewer recognizes the importance of “*an experimental demonstration of practically useful current-induced switching of antiferromagnetic order*”. As pointed out by the reviewer, most of the previous results focused on the current-induced switching of biaxial AFMs, either by Néel SOTs [R12, R13] or from interfacial SOTs using AFM/HM structures [R5, R14, R15]. Note that most of these studies utilized epitaxial films lattice-matched to the underlying substrate, posing a challenge to integrate with the existing Si-based complementary metal-oxide semiconductor (CMOS) circuits. Besides, the resistively switched states also showed signatures of relaxation [R5], detrimental for non-volatile memory applications using AFMs. In contrast, our work employs CMOS-compatible metallic polycrystalline AFMs which has been previously used and optimized for pinning layers of spin valve structures [R16]. One key aspect of our work is the demonstration of current-induced switching of antiferromagnetic Néel vector in a CMOS-friendly AFM metal with long-term data retention capabilities, suitable for practical utilization of antiferromagnetic spintronics.

To highlight this point, we have added the following sentence in our manuscript as follows:

<page 17, line 351>

Contrary to the common understanding that bi-axial AFMs are required for electrical switching, our results demonstrate the potential of polycrystalline antiferromagnetic metals compatible with existing complementary metal-oxide semiconductor (CMOS) technology for low-current operation and high thermal stability antiferromagnetic spintronics.

Next, we address the explanation for the thresholds in PtMn/Pt system. We speculate that the influence of easy-plane anisotropy contribution [R1-R3] and interfacial chiral DMI [R3, R4] results in the stabilization of inhomogeneous multi-domain AFM ground state in our PtMn/Pt structures. The injection of current in the Pt layer results in bulk or interfacial spin-orbit torques (SOTs) from spin-Hall effect which can act on the Néel DWs [R3, R5] and/or antiferromagnetic topological spin textures [R4, R6-R8]. In this scenario, the reversible resistive states can be attributed to the 90° rotation of the in-plane projection of the antiferromagnetic Néel vector, consistent with previous micromagnetic simulations [R3, R7] and some experimental results [R6, R7, R8]. The crystallographic or magnetic defects and grain boundaries in our polycrystalline PtMn can potentially act as pinning centers, impeding the thermally activated SOT-induced rearrangement or motion of the Néel DWs [R5] and nucleation or motion of spin textures such as vortex-antivortex pairs [R7, R8], skyrmions or bimerons [R4]. Thus, a possible situation for the threshold feature (Fig. 3(a) of the revised manuscript) represents a cross-over between different dynamical regimes (*e.g.* depinning, flow etc.) of current-induced motion of antiferromagnetic spin textures, similar to the ferromagnetic case [R17, R18]. Future theoretical and/or micromagnetic modeling concerning SOT-induced thermally activated motion of antiferromagnetic DWs or topological spin structures in the presence of pinning are expected to shed light on the observed experimental results. Accordingly, we have added the following sentence in our manuscript:

<page 17, line 347>

In principle, the threshold-like behaviour observed in ΔR_{Hall} versus J_{Pt} , can be attributed to the cross-over between various regimes of spin texture dynamics characterized by different pinning strengths.

To address the issue regarding the effect of Joule heating and switching amplitude (R_{Hall}), we separately examine the effect of various factors arising from the application of current in PtMn/Ru structures. Note that while our unipolar saw-tooth (triangular) switching characteristics are similar to some of the previous results [R19, R20], the current-induced torques between PtMn and these systems are likely to be different. For PtMn/Ru, under the assumption of an inhomogeneous multi-domain ground state, the application of current can lead to a spin-transfer torques (STTs) acting on the antiferromagnetic textures [R5, R10], thermal reorientation of the antiferromagnetic Néel vector [R19], and/or electromigration [R20]. Comparison of our experimental results with previous studies [R21] indicates a similar range of current densities (J_{PtMn}) resulting in observable saw-tooth switching characteristics in PtMn/Ru to that required for STT-induced DW motion in ferrimagnetic structures. Besides, we also observe a strong current pulse width dependence and reduction of R_{Hall} for successive cycles (Fig. 2(b) of the revised manuscript), qualitatively similar to some previous studies on metallic AFMs [R19], indicating the presence of thermal effects. In addition, we also observe possible electromigration effects manifesting as darkened regions for PtMn/Ru after several switching cycles (see supplementary Fig. S10). While our experiments cannot separately distinguish these factors, we associate the switching characteristics to a combination of thermally assisted

reorientation of antiferromagnetic textures along with possible contributions from STT and other non-magnetic effects.

Accordingly, we have revised the manuscript as follows:

From:

<page 13, line 268>

In accordance with previous Monte-Carlo simulations of thermal effects on switching in AFMs¹⁹, the observed triangular switching characteristics and reduction of ΔR_{Hall} versus $1/t_{\text{PtMn}}$ in these structures could indicate thermally-activated switching dynamics⁴⁴.

To:

<page 16, line 309>

For PtMn/Ru, the current flowing through the AFM layer can also result in STTs acting on the multi-domain ground state. Theoretical calculations have shown that adiabatic and non-adiabatic components of STTs can lead to translational motion of AFM DWs^{24,55}, which could result in a magnetoresistive response owing to the finite magnetization in the DW region. For this scenario, the threshold current required to induce DW motion should remain unchanged with increasing t_{PtMn} , roughly consistent with our observed results (Fig. 3(b)). However, the non-saturating behaviour of switching amplitude and its variation in successive cycles indicate intermingled contributions from other sources such as thermally-activated reorientation of antiferromagnetic Néel vector^{20,39} and/or non-magnetic contributions arising from electromigration⁴⁰. Whereas our electrical measurements do not enable us to distinguish these factors, the present observations are likely related to a combination of thermally-activated reorientation of the antiferromagnetic Néel vector with possible contributions from STT and other nonmagnetic effects.

Finally, let us discuss the significance of this work from engineering perspective. Actually, we have no intention to claim the superiority of the present metallic AFM to the previously studied biaxial systems [R5, R12-R15] in terms of switching properties. The key result of this work is the demonstration of current-induced manipulation of antiferromagnetic Néel vector in PtMn with long-term data retention capabilities and compatibility with the existing Si technology as described above. We would like to also emphasize that the partial switching widely observed in antiferromagnetic systems is not necessarily a detrimental behavior, because such a phenomenon can be useful for neuromorphic computing [R22, R23]. Thus, we believe that the observation of switching in PtMn is highly valuable for the future of antiferromagnetic spintronics.

Reviewer's Comment:

Some minor questions:

1. It seems that Ru(1 nm) is used as a capping layer of PtMn/Ru. Does it free from oxidation when the sample is exposed to air? If oxidized, does it affect the spin-orbit torque?

Our Reply:

We believe that the capping layer of Ru (1 nm) is almost oxidized. The evaluation of current flow ratios from sheet resistance measurements indicate that there is negligible current flowing through the capping layer. Owing to this, we do not consider SOTs originating from current flow in the capping layer.

Accordingly, we have added the following sentence in the revised manuscript:

<page 15, line 296>

Owing to the negligible current flow through the oxidized capping layer, we do not consider its contribution for both PtMn/Pt and PtMn/Ru structures.

Reviewer's Comment:

2. In Fig. 1(b), the magnetic moment of PtMn/Pt is small but clearly finite. Where does it come from?

Our Reply:

There are two possibilities for the origin of the finite magnetic moment in our PtMn/Pt structures. It can either arise from (i) minute fractions of disordered moment in bulk or interface of the AFM PtMn, and/or (ii) inhomogeneous multi-domain ground state of the AFM under the competition of exchange, anisotropy and interfacial DMI. Separate X-ray circular magnetic dichroism (XMCD) measurements at the Mn L_3 edge on AFM/ferromagnet PtMn/[Co/Ni] multilayer structures do not reveal any discernable XMCD signals upon the application of current, ruling out any dominant contribution of these moments towards the observed current-induced switching behavior.

Accordingly, we have added the following sentences in the revised manuscript:

<page 5, line 106>

The finite m can possibly arise from minute fractions of disordered moments and/or inhomogeneous multi-domain antiferromagnetic configuration (shown later).

And,

<page 11, line 225>

Separate X-ray circular magnetic dichroism (XMCD) measurements at Mn L_3 edge on AFM/FM PtMn/[Co/Ni] multilayer structures do not reveal any discernable XMCD signals upon the application of current, ruling out any dominant contribution from disordered or uncompensated moments.

Reviewer's Comment:

3. What is the easy axis of PtMn? Is it normal to the plane?

Our Reply:

We thank the reviewer for his/her question. The magnetic easy-axis for polycrystalline PtMn thin films is an unresolved issue and is known to be critically sensitive to temperature [R24], chemical substitution/doping [R1, R9] and strain from the underlying layers or substrate [R2]. Polarized neutron diffraction experiments on epitaxial MgO/PtMn thin films have indicated that the Mn moments are aligned perpendicular to (001) direction and tilted at 45° to (100) direction [R25]. Moreover, these results are different from the previous studies on bulk which showed two possible easy axes, along (001) and (100) crystallographic directions [R25, R26]. Besides, the possibility of an inhomogeneous multi-domain configuration for polycrystalline thin films might cause different magnetic orientations among adjacent crystal grains. Our experimental results from XMLD investigations show the presence of alternating dark and bright contrast areas in LV and LH configurations, indicating a coexistence of mixed out-of-plane and in-plane antiferromagnetic Néel vector orientation and easy axes. We believe that future polarized neutron diffraction studies or X-ray magnetic imaging on polycrystalline PtMn thin films could provide quantitative insights to the magnetic easy axis of PtMn.

Accordingly, we have added the following portion in the discussion part of the manuscript:

<page 14, line 276>

Our sputtered films, however, have a polycrystalline structure. Previous studies pointed out that significant magnetostriction coefficient⁴³ and the sensitivity of Mn-based AFMs to crystallinity and/or chemical composition^{28,29,31,44} (e.g. effects of Mn substitution/doping and valence electron number) could induce an easy-plane magnetic anisotropy⁴⁵⁻⁴⁷, resulting in multiple stable Néel vector orientations in the polycrystalline films. Note that our observation of a reversible XMLD contrast along LV and LH configurations (Figs. 4(e)-(k)) is also consistent with the above scenario, indicating the possibility of both out-of-plane and in-plane Néel vector components.

Reviewer #3 (Remarks to Author):

Reviewer's Comment:

The manuscript describes a spin orbit torque induced magnetization switching in antiferromagnet PtMn layer at room temperature and the PtMn is uniaxial. The work is potentially interesting and present that a uniaxial antiferromagnets could also be switched via spin orbit torque. They also provide XMLD-PEEM images to support the AFM moments switching in PtMn.

Based on the above, I feel that the work described fits within the scope of Nature Communications.

Our Reply:

We are pleased to learn that the reviewer finds our work interesting and is supportive of the publication of our manuscript.

Reviewer's Comment:

Nevertheless, I find some of the explanations a bit obscure and the following points must be addressed.

1. My main issue is the justification of magnetic anisotropy of PtMn. The authors mentioned the bulk PtMn occupy a uniaxial anisotropy. However, the PtMn in this manuscript is polycrystalline, which indicates the PtMn might not be uniaxial anymore. Could the author show the XRD spectra of PtMn? Whether the SOT switching could happen in highly textured PtMn?

Our Reply

We thank the reviewer for his/her question. The magnetic easy-axis and anisotropy of polycrystalline PtMn thin films is an open question which is known to be critically sensitive to various factors such as temperature [R24], chemical substitution/doping [R1, R9] and strain from the underlying layers or substrate [R2]. First principles calculations on bulk PtMn indicated a uniaxial magnetic anisotropy [R1], where the Mn moments are oriented parallel to the (001) direction [R26]. Subsequent calculations suggested a significant modification of the bulk uniaxial component and stabilization of a partial or dominant easy-plane contribution associated with substrate-induced strain effects [R2] and/or chemical composition [R1, R9]. In addition, starting from a uniaxial spin configuration with the Mn moments along (001) axis, the stabilization of an easy-plane magnetic anisotropy is also associated with a rotation of the antiferromagnetic Néel vector along other crystallographic directions [R2]. Thus, the bulk uniaxial nature could be partially or completely modified into an easy-plane magnetocrystalline anisotropy, and a combination of antiferromagnetic exchange, anisotropy, and interfacial DMI can result in an inhomogeneous multi-domain antiferromagnetic ground state configuration in polycrystalline PtMn films [R3, R4].

We also performed an out-of-plane ($\theta - 2\theta$) X-ray diffraction (XRD) measurement to clarify the crystalline orientation of PtMn layer using Ta(3)/Pt(3)/MgO(2)/PtMn(10)/Ru(2) structure. Figure R2 shows the XRD spectra of the stacks before and after annealing. Only reflections from face-centered cubic (fcc) (or face-

centered tetragonal (fct)-PtMn(111), fcc-Pt(111) are observed, indicating that PtMn and Pt layers have (111) orientation of fcc (or $L1_0$ -ordered fct for PtMn) phase. Similar spectra are observed irrespective of PtMn thickness. In addition, the shift of the (111) peak position from 39.8° (for as-deposited state) to 40.2° (for annealed state) is expected to originate from the phase change of PtMn from the disordered fcc to $L1_0$ -ordered tetragonal structures [R26]. The XRD patterns for as-deposited and annealed PtMn thin films have been newly added in supplementary information S1 of the revised manuscript.

Figure R2: Out-of-plane X-ray diffraction (XRD) patterns for Ta(3)/Pt(3)/MgO(2)/PtMn(10)/Ru(2) heterostructures in as-deposited and annealed state, respectively. Annealing conditions are the same as mentioned in the main text.

Next, we address the issue regarding SOT-switching in textured PtMn thin films. We speculate that the concerted action of antiferromagnetic exchange, anisotropy [R1-R3] and interfacial DMI [R3, R4] results in the stabilization of inhomogeneous multi-domain antiferromagnet (AFM) ground state with Néel domain walls (DWs) [R3, R5] and/or other topological spin textures [R4, R6-R8], whose chirality is fixed by the sign and magnitude of D . Note that an estimate of D for PtMn/Pt structure, using experimentally obtained magnitudes of domain size and typical values of magnetic parameters, is comparable to the critical D_C , required for the stabilization of an inhomogeneous multi-domain state. A detailed discussion and estimates of DW width, D and D_C using micromagnetic parameters typical for Mn-based metallic AFMs have been newly added in supplementary information S7 and S8 of the revised manuscript. Under this scenario, the damping-like and field-like components of SOTs, originating from the injection of current in the Pt layer, are capable of reorientation of antiferromagnetic Néel vector. The proposed scenario is also consistent with previous micromagnetic simulations [R3, R8] demonstrating reversible 90° rotation of the antiferromagnetic Néel vector under the action of damping-like SOTs, either by domain reorientation (for $D < D_C$) and/or motion of DWs or nucleation of topological spin textures (for $D > D_C$).

Accordingly, we have added the following sentences in the revised manuscript:

<page 16, line 329>

As stated before, a combination of uniaxial and easy-plane anisotropies along with interfacial DMI can lead to the spontaneous multi-domain configuration comprising Néel DWs and/or topological spin textures^{46,47,51-53} in PtMn/Pt structures. In fact, these predictions have been confirmed by recent experiments demonstrating imprinted antiferromagnetic vortex states on an adjacent ferromagnetic layer in AFM/FM^{51,52} or exotic topological meron-antimeron pairs in AFM/HM structures⁵³. The twisting of the antiferromagnetic Néel vector around these spin textures leads to non-zero Néel topological charge, endowing protected spin configurations with distinct magnetic polarities and chiralities (sizes ~ hundreds of nm) and robust thermal stability. Besides, numerical simulations also suggest efficient nucleation and motion of these antiferromagnetic spin textures under the action of SOTs in AFM/HM⁵⁷. Thus, a possible scenario concerning the polarity dependent current-induced switching characteristics is attributed to the action of SOTs on the inhomogeneous multi-domain configuration.

Reviewer's Comment:

2. The AFM moments dynamics is not clear in Fig.4. Whether the AFM moments are switched perpendicular to the spin polarization direction(parallel to current)?

Our reply:

We thank the reviewer for his/her question. As stated above, we speculate that observed switching characteristics arises from a rotation of the average in-plane component of antiferromagnetic Néel vector where both the initial and final antiferromagnetic states are inhomogeneous and multi-domain in nature. The switching between inhomogeneous configurations is dominated by the dynamics of DWs rather than the antiferromagnetic domains themselves, as pointed out in previous micromagnetic simulations using typical parameters resembling PtMn/Pt structures [R3]. Thus, the action of SOTs orients the antiferromagnetic DW moments parallel to the polarization of incoming spin-current, similar to the dynamics of SOT-induced motion of ferromagnetic DWs [R16]. However, we would also like to point out that while the above picture can explain the occurrence of high and low resistive states, it cannot fully account for the observed current polarity dependence of switching in PtMn/Pt. We speculate that the polarity dependent characteristics arises from possible contributions of nucleation or motion of other topological antiferromagnetic spin textures such as vortex-antivortex pairs [R8], skyrmions or bimerons [R4, R6, R7]. Future theoretical or micromagnetic modelling considering polycrystalline AFM/HM structures with parameters mimicking experimentally obtained results can provide significant insights into the dynamics of AFM moments in these practically applicable metallic structures.

Accordingly, we have added the following sentence in the revised manuscript:

<page 17, line 341>

Irrespective of the initial multi-domain configuration, the switching dynamics is expected to proceed via a 90° rotation of the in-plane projection of Néel vector towards the spin-polarization direction⁴⁷ by rearrangement or motion of DWs, and/or current-induced nucleation and motion of vortex-antivortex pairs, skyrmions, or bimerons^{53,57}.

Reviewer's Comment:

3. The switching behavior in PtMn doesn't resemble the characteristics of Mn₂Au as it is nearly saturated in Mn₂Au cases.

Our reply:

We thank the reviewer for his/her comment. Accordingly, we have revised the manuscript as:

From:

<page 8, line 165>

This triangular switching behaviour also persists for various τ_p and t_{PtMn} (see supplementary Fig. S4(b)-(d)), and closely resembles the switching characteristics observed in previous works on Mn₂Au^{17,19-21}.

To:

<page 10, line 186>

This sawtooth-like nature also persists for various τ_p and t_{PtMn} (see supplementary Fig. S4(b)-(d)), and closely resembles the switching characteristics observed in some previous works^{39,40}.

Reviewer's Comment:

(4) The authors observe a current-polarity dependent switching behaviors. I'd like to see more discussions on this issue.

Our Reply:

We thank the reviewer for his/her suggestion. Based on this suggestion, we have added the following sentences in our revised manuscript:

<page 17, line 339>

Thus, a possible scenario concerning the polarity dependent current-induced switching characteristics is attributed to the action of SOTs on the inhomogeneous multi-domain configuration. Irrespective of the initial multi-domain configuration, the switching dynamics is expected to proceed via a 90° rotation of the in-plane projection of Néel vector towards the spin-polarization direction⁴⁷ by rearrangement or motion of DWs, and/or current-induced nucleation and motion of vortex-antivortex pairs, skyrmions, or bimerons^{53,57}.

References:

- [R1] Umetsu, R. Y., Sakuma, A. & Fukamichi, K. Magnetic anisotropy energy of antiferromagnetic L10-type equiatomic Mn alloys. *Appl. Phys. Lett.* **89**, 052504 (2006).
- [R2] Park, I. J. *et al.* Strain control of the Néel vector in Mn-based antiferromagnets, *Appl. Phys. Lett.* **115**, 142403 (2019).
- [R3] Tomasello, R. *et al.* Domain periodicity in an easy-plane antiferromagnet with Dzyaloshinskii-Moriya interaction. [arXiv:2004.01944](https://arxiv.org/abs/2004.01944) (2020).
- [R4] Jani, H. *et al.* Half-skyrmions and Bimerons in an antiferromagnetic insulator at room temperature. [arXiv:2006.12699v](https://arxiv.org/abs/2006.12699v) (2020).
- [R5] Baldrati, L. *et al.* Mechanism of Néel order switching in antiferromagnetic thin films revealed by magnetotransport and imaging techniques, *Phys. Rev. Lett.* **123**, 177201 (2019).
- [R6] Wu, J. *et al.* Direct observation of imprinted antiferromagnetic vortex states in CoO/Fe/Ag(001) discs. *Nature Phys.* **7**, 303 (2011).
- [R7] Chmiel, F. P. *et al.* Observation of magnetic vortex pairs at room temperature in a planar α -Fe₂O₃/Co heterostructure. *Nature Mater.* **17**, 581 (2018).
- [R8] Shi, J. *et al.* Electrical manipulation of the magnetic order in antiferromagnetic PtMn pillars. *Nature Elec.* **3**, 92 (2020).
- [R9] Andresen, A., Kjekshus, A., Møllerud, R. & Pearson, W. B. Equiatomic transition metal alloys of manganese IV. A neutron diffraction study of magnetic ordering in the PtMn phase. *Philos. Mag.: A Journal of Theoretical Experimental and Applied Physics.* **11**, 1245 (1965).
- [R10] Xu, Y., Wang, S. & Xia, K. Spin-Transfer Torques in Antiferromagnetic Metals from First Principles. *Phys. Rev. Lett.* **100**, 226602 (2008).
- [R11] Tveten, E., G., Qaiumzadeh, A. Tretiakov, O. A. and Brataas, A. Staggered Dynamics in Antiferromagnetic by Collective Coordinates. *Phys. Rev. Lett.* **110**, 127208 (2013).
- [R12] Wadley, P. *et al.* Electrical switching of an antiferromagnet. *Science* **351**, 587-590 (2016).
- [R13] Bodnar, S. Y. *et al.* Writing and reading of antiferromagnetic Mn₂Au by Néel spin-orbit torques and large anisotropic magnetoresistance. *Nature Commun.* **9**, 348 (2018).
- [R14] Chen, X. Z. *et al.* Antidamping-Torque-Induced Switching in Biaxial Antiferromagnetic Insulators. *Phys. Rev. Lett.* **120**, 207204 (2018).
- [R15] Moriyama, T., Oda, K., Ohkochi, T., Kimata, M. & Ono, T. Spin torque control of antiferromagnetic moments in NiO. *Sci. Rep.* **8**, 14167 (2018).
- [R16] Nozières, J. P. Blocking temperature distribution and long-term stability of spin-valve structures with Mn-based antiferromagnets. *J. Appl. Phys.* **87**, 3920 (2000).
- [R17] Thiaville, A. Rohart, S., Jué, É., Cros, V. and Fert A. Dynamics of Dzyaloshinskii domain walls in ultrathin magnetic films. *Europhys. Lett.* **100**, 57002 (2012).

- [R18] Montoya, S. A. et al., Spin-orbit torque induced dipole skyrmion motion at room temperature. *Phys. Rev. B.* **98**, 104432 (2018).
- [R19] Meinert, M., Graulich, D. & Mallata-Wagner, T. Electrical Switching of Antiferromagnetic Mn₂Au and the Role of Thermal Activation. *Phys. Rev. Appl.* **9**, 064040 (2018).
- [R20] Cheng, Y., Yu, S., Hwang, J. & Yang, F. Electrical Switching of Tristate Antiferromagnetic Néel Order in α -Fe₂O₃ Epitaxial Films. *Phys. Rev. Lett.* **124**, 027202 (2020).
- [R21] Okuno, T. *et al.* Spin-transfer torques for domain wall motion in antiferromagnetically coupled ferrimagnets, *Nature Elec.* **2**, 389 (2019).
- [R22] Borders, W. A. *et al.* Analogue spin-orbit torque device for artificial-neural-network-based associative memory operation. *Appl. Phys. Express* **10**, 013007 (2017).
- [R23] Kurenkov, A. *et al.* Artificial Neuron and Synapse Realized in an Antiferromagnet/Ferromagnet Heterostructure using Dynamics of Spin-Orbit Torque Switching. *Adv. Mater.* **31**, 1900636 (2019).
- [R24] Hama H., Motoruma, R., Shinozaki, T. & Tsunoda, Y. Spin-flip transition of L10-type MnPt alloy single crystal studied by neutron scattering, *J. Phys.: Condens. Matter* **19**, 176228 (2007).
- [R25] Solina, D. *et al.* The magnetic structure of L10 ordered MnPt at room temperature determined using polarized neutron diffraction. *Mater. Res. Express* **6**, 076105 (2019).
- [R26] Krén, E. *et al.* Magnetic Structures and Exchange Interactions in the Mn-Pt System. *Phys. Rev.* **171**, 574 (1968).
- [R27] Ladwig, P. F. *et al.* Paramagnetic to antiferromagnetic phase transformation in sputter deposited Pt-Mn thin films, *J. Appl. Phys.* **94**, 979 (2003).

Reviewers' Comments:

Reviewer #1:

None

Reviewer #2:

Remarks to the Author:

The authors' responses are satisfactory. I support publication of this work.

Reviewer #3:

The authors have provided a comprehensive response to questions of all three referees. Their experimental results are very interesting, so I recommend publication of their manuscript.

Page 2, line 5 from the bottom about previous reports on the field-like Néel SOT, the author could add another recent demonstration of electric-field control of Néel SOT.

“Chen X. Z. *et al.*, Electric field control of Néel spin orbit torque in an antiferromagnet. *Nat. Mater.* 18, 931 (2019) ”

Response to reviewer's comments

Reviewer #2 (Remarks to the author)

Reviewer's Comment

The author's responses are satisfactory. I support publication of this work.

Our reply

We are pleased to learn that the reviewer is satisfied with our response and support the publication of our manuscript.

Reviewer #3 (Remarks to the author)

Reviewer's Comment

The authors have provided a comprehensive response to questions of all three referees. Their experimental results are very interesting, so I recommend publication of their manuscript.

Our reply

We are pleased to learn that the reviewer finds our work interesting and recommends publication of our manuscript.

Reviewer's Comment

Page 2, line 5 from the bottom about previous reports on the field-like Néel SOT, the author could add another recent demonstration of electric-field control of Néel SOT.

“Chen X.Z. et al., Electric field control of Néel spin orbit torque in an antiferromagnet. Nat. Mater. 18, 931 (2019)”

Our reply

We thank the reviewer for his/her suggestion. The revised portion to respond to this issue is as follows:

<page 2, line 48>

Switching of an AFM either by field-like Néel SOTs originating from inverse spin galvanic effects^{7,18-21} or SOTs in AFM-insulator/heavy metal (HM) heterostructures²²⁻²⁴ and electric field control of Néel SOTs²⁵ have been demonstrated, offering techniques to manipulate antiferromagnetic Néel vector.

The reference is added as reference number 25 of the revised version of the main manuscript as follows:

<page 21, line 467>

25. Chen, X. *et al.* Electric field control of Néel spin-orbit torque in an antiferromagnet. *Nature Mater.* **18**, 931 (2019).

In addition, we have also added another reference in the revised version of the main manuscript as follows:

<page 24, line 539>

59. Fujii, J. *et al.* Evidence for in-plane spin-flop orientation at the MnPt/Fe (100) interface revealed by x-ray magnetic linear dichroism, *Phys. Rev. B.* **73**, 214444 (2006).